# microRNAs and Other Serological Markers of Liver Fibrosis in Patients with Alcohol-Related Liver Cirrhosis

**DOI:** 10.3390/biomedicines12092108

**Published:** 2024-09-16

**Authors:** Agata Michalak, Małgorzata Guz, Joanna Kozicka, Marek Cybulski, Witold Jeleniewicz, Karolina Szczygieł, Ewa Tywanek, Halina Cichoż-Lach

**Affiliations:** 1Department of Gastroenterology, Medical University of Lublin, Jaczewskiego 8, 20-954 Lublin, Poland; joanna.kozicka@umlub.pl (J.K.); halina.lach@umlub.pl (H.C.-L.); 2Department of Biochemistry and Molecular Biology, Medical University of Lublin, Chodźki 1, 20-093 Lublin, Poland; malgorzata.guz@umlub.pl (M.G.); marek.cybulski@umlub.pl (M.C.); witold.jeleniewicz@umlub.pl (W.J.); 3Clinical Dietetics Unit, Department of Bioanalytics, Medical University of Lublin, Chodźki 7, 20-093 Lublin, Poland; karolina.szczygiel@umlub.pl; 4Department of Internal Medicine and Internal Medicine in Nursing, Medical University of Lublin, Chodźki 7, 20-093 Lublin, Poland; ewa.tywanek@umlub.pl; 5Department of Endocrinology with Nuclear Medicine Department, Center of Oncology of the Lublin Region St. Jana z Dukli, Jaczewskiego 7, 20-090 Lublin, Poland

**Keywords:** microRNAs, liver cirrhosis, markers of liver fibrosis, alcoholic liver disease

## Abstract

**Background**: It is essential to identify novel non-invasive markers of liver fibrosis for clinical and scientific purposes. Thus, the goal of our survey was to assess the serological expression of selected microRNAs (miRNAs) in patients with alcohol-related liver cirrhosis (ALC) and to correlate them with other existing markers. **Methods**: Two hundred and thirty-nine persons were enrolled in the study: one hundred and thirty-nine with ALC and one hundred healthy controls. Serological expression of miR-126-3p, miR-197-3p and miR-1-3p was evaluated in all participants. Direct markers of liver fibrosis (PICP, PIIINP, PDGF-AB, TGF-α and laminin) together with indirect indices (AAR, APRI, FIB-4 and GPR) were also assessed. The additional evaluation concerned hematological parameters: MPV, PDW, PCT, RDW, MPR, RPR NLR, PLR and RLR. **Results**: The expression of miR-197-3p was lower in ALC compared to controls (*p* < 0.0001). miR-126-3p correlated negatively with AST (*p* < 0.05) and positively with miR-197-3p (*p* < 0.001). miR-197-3p correlated with direct markers of liver fibrosis—positively with PDGF-AB (*p* < 0.005) and negatively with TGF-α (*p* < 0.01). Significant negative relationships were noticed between miR-1-3p and the number of neutrophils (*p* < 0.05), TGF-α (*p* < 0.05) and laminin (*p* < 0.05). **Conclusions**: The achieved results and observed correlations prove the potential involvement of the examined miRNAs in the process of liver fibrosis, giving a novel insight into the diagnostics of liver cirrhosis.

## 1. Introduction

Liver cirrhosis (LC), together with other chronic liver diseases, permanently constitutes an alarming global medical problem, leading to significantly impaired quality of life and ranking as the 14th most common cause of death globally. Therefore, the diagnosis of liver fibrosis, possibly at the earliest stage, is an obvious therapeutic strategy. Liver biopsy still remains a gold standard here, but due to its limitations, the role of non-invasive tests in this field (serum biomarkers and/or imaging tools) is increasing. These non-invasive modalities are safer, less expensive and easier to be repeated in the course of the disease to monitor its progress. Thus, from both scientific and clinical points of view, it is worth identifying novel non-invasive markers of liver fibrosis, especially serological ones, due to their convenience in clinical settings [1,2,3,4]. Following the results of research on the role of microRNAs (miRNAs) in various liver pathologies, it is easy to conclude that there is basically no disease in which a specific miRNA family is not involved [5,6]. However, miRNAs are relatively rarely compared to other markers of liver function to determine their potential relationships [7,8,9,10]. Therefore, it is hard to place them accurately among common markers used in the course of liver disorders. Perhaps such trials would allow for the introduction of miRNAs into everyday diagnostics in hepatology to an even greater extent. At the same time, the multitude of already known miRNA types makes their selection for subsequent research a great challenge. However, there are still miRNA molecules that have rarely been evaluated in patients with liver disease. Such conclusions can be drawn by analyzing the available literature on miR-1-3p, miR-126-3p and miR-197-3p. It seems that the mentioned particles have not yet been tested comprehensively in the group of patients with alcohol-related liver cirrhosis (ALC). The published data on these miRNAs are mainly concerned with their role in the development of liver cancer or oxidative stress among hepatocytes. They appear to be not investigated in cirrhotic patients to assess their relationship with liver fibrosis. Moreover, no clinically significant dependences between miR-1-3p, miR-126-3p, miR-197-3p and both direct and indirect parameters of liver fibrosis have been identified so far. These observations constituted the major rationale for conducting a current survey. Furthermore, the lack of commonly accessible data on potential correlations between miRNAs and serological markers of liver fibrosis appears to be the gap in hepatology that should be filled. Indirect and direct serum indicators have been known for years in the diagnosis/monitoring of liver fibrosis. In selected clinical situations, they may constitute an alternative to liver biopsy and non-invasive methods of imaging the advancement of liver fibrosis, and in everyday clinical practice, they act as predictors among patients with chronic liver pathologies [11,12,13,14]. Indirect markers enable the verification of liver function disorders and are the determinants of its efficiency (e.g., liver enzymes, bilirubin, blood coagulation parameters and proteins physiologically synthesized by hepatocytes). However, from a diagnostic point of view, they are not perfect, and further assessment of cirrhotic patients usually still requires the assessment of the severity of liver fibrosis with imaging studies or liver biopsy. Simultaneously, due to the inflammatory character of liver fibrosis, new players in this field have approached, namely hematological indices. Nonetheless, all the above-mentioned parameters are already known, and potential novel non-invasive markers of liver fibrosis are required. Therefore, the primary goal of the current manuscript was to evaluate the potential role of selected miRNAs in the diagnostics of ALC. Additionally, we looked for significant relationships between miRNAs and commonly used indices of liver fibrosis.

## 2. Materials and Methods

Two hundred thirty-nine people were recruited for the study: one hundred and thirty-nine patients with ALC and one hundred healthy people (non-drinkers reporting only occasional alcohol consumption of no more than 10 g of pure alcohol per day or complete abstinence, with no liver disorders or other chronic comorbidities) as part of the control group. The research group included patients of the Gastroenterology Department with the Endoscopy Unit of the Medical University of Lublin and the Gastroenterology Outpatient Clinic included in the study in 2017–2020. The control group consisted of healthy volunteers. The clinical description of the study participants is presented in Table 1, and the biochemical description, along with the scores of the scales used, is shown in Table 2.

The following inclusion criteria were applied in the study group:The diagnosis of ALC was based on the presence of nodular liver remodeling on imaging studies and on a history of alcohol abuse.Nodular liver remodeling was diagnosed by a liver ultrasound examination.Doppler ultrasound examination of the hepatic portal system was used to confirm the presence of hepatic portal hypertension (portal vein diameter ≥ 13 mm) and to exclude its other causes.Alcohol-related LC was diagnosed in the event of a confirmed daily consumption of more than 20 g of pure alcohol in the case of women and 30 g—in the case of men. Information regarding alcohol history was obtained from patients or their family members.Additionally, all examined people from this group presented a positive result of the screening test AUDIT-C (The Alcohol Use Disorders Identification Test Consumption). This test includes the following 3 questions: How often do you drink drinks containing alcohol? How many servings of alcohol (1 serving is approximately 10 g of 100% alcohol) do you drink on average when you drink alcohol? How often do you drink six or more drinks at one time? Obtaining 3 points in the case of women and 4 in the case of men in the total score of the AUDIT-C test (maximum 12 points to be obtained) indicates the existence of an alcohol problem (harmful drinking or addiction).Exclusion criteria in the study group concerned were as follows:The diagnosis of spontaneous bacterial peritonitis—≥ 250 NEU/µL of ascitic fluid/positive fluid culture result.The presence of hepatic encephalopathy. The CHESS (Clinical Hepatic Encephalopathy Staging Scale) scale was used to exclude it. Each study participant obtained a score of 0/9 points.Patients with hepatorenal syndrome type 1 (no increase in creatinine concentration ≥0.3 mg/dL within 48 h or ≥50% within 7 days of hospitalization) and type 2 (creatinine concentration among the study subjects was <1.5 mg/dL).The exclusion criteria in the research and control groups were as follows:Other potential etiological factors of liver diseases, including viral and autoimmune pathologies. Serological indicators for hepatitis A/B/C virus, Epstein–Barr virus and cytomegalovirus were negative. The presence of antinuclear antibodies, antimitochondrial antibodies, anti-smooth muscle antibodies and antibodies against liver-kidney microsomes was excluded.The presence of clinically significant acute or chronic inflammation.The abnormal result of serological alkaline phosphatase.Ongoing therapy with steroids or antibiotics at baseline or within the last three months.

Serological expression of miR-126-3p, miR-197-3p and miR-1-3p was evaluated in all examined patients (miREIA; BioVendor, Brno, Czech Republic). Direct markers of liver fibrosis [procollagen I carboxyterminal propeptide (PICP), procollagen III aminoterminal propeptide (PIIINP), platelet-derived growth factor AB (PDGF-AB), transforming growth factor-α (TGF-α) and laminin] together with indirect indices (AAR, APRI, FIB-4 and GPR) were also assessed. To obtain the results of direct indices, ELISA kits were applied [PICP, PIIINP—Wuhan EIAab Science (Wuhan, China), PDGF-AB, TGF-α—R&D Systems Quantikine ELISA Kit (Minneapolis, MN, USA), laminin—EIA Kit without Sulphuric Acid (Kusatsu, Shiga, Japan). The evaluation of hematological parameters included mean platelet volume (MPV), platelet distribution width (PDW), plateletcrit (PCT), red blood cell distribution width (RDW), MPV-to-platelet (PLT) ratio (MPR), RDW-to-PLT ratio (RPR), neutrophil-to-lymphocyte (LYM) ratio (NLR), PLT-to-LYM ratio (PLR) and RDW-to-LYM ratio (RLR).

Statistical analysis was performed with Statistica version 13.3 (TIBCO Software Inc.; Santa Clara, CA, USA) software for Windows. Deviation from normality was evaluated using the Kolmogorov–Smirnov test. The Mann–Whitney U test was applied for between-group comparisons because of non-normal distribution. Spearman correlation analyses were performed to verify the correlations. All probability values were two-tailed, and a value of *p* less than 0.05 was considered statistically significant. The positive predictive value (PPV) and negative predictive value (NPV) of the studied indicators were also determined. A significance level of *p* < 0.05 was considered statistically significant.

Receiver operating characteristic (ROC) curves and area under the curve (AUC) values were used to assess the sensitivity, specificity and proposed cut-offs of investigated miRNAs in ALC patients. ROC analysis was performed using Medical Bundle for Statistica software. Optimal cut-off values for miRNA levels were determined using the tangent method.

Each participant gave written consent to participate in the survey. The study was carried out in accordance with the protocol approved by the Bioethics Committee of the Medical University of Lublin (decision no. KE-0254/86/2016).

## 3. Results

The results of the tested miRNA molecules (miR-126-3p, miR-197-3p and miR-1-3p) are presented in Table 3.

The serum concentration of miR-197-3p in ALC patients was significantly lower compared to the control group (*p* < 0.0001). The expression of both miR-126-3p and miR-1-3p did not differ notably between research and control groups. The results of the evaluated hematological parameters in the research and control groups are presented in Table 4.

The medians of PLT and LYM in patients with ALC were below the reference range. The median MPV in patients with ALC remained within the reference values. The medians of PDW and PCT among the examined subjects were within the reference values. An increase in the median RDW and NEU above the reference values was demonstrated in the ALC group. Of the hematological parameters examined in the ALC group, only the median of MPV did not differ significantly compared to the control group; the medians of MPR, PDW, RDW, RPR, NEU and NLR were significantly higher, and PLT, PCT, LYM and PLR were lower than in the control group. The analysis of AAR, APRI, FIB-4 and GPR indicators showed their notably higher medians in patients with ALC compared to the control group. The results of the examined indirect parameters of liver fibrosis in the studied groups are presented in Table 5.

Among patients with ALC, the medians of direct markers of liver fibrosis (except PICP and laminin) were significantly lower. The medians of PICP and laminin did not differ notably. The research results are illustrated in Table 6.

Correlations between the examined miRNA molecules and other determined indicators were assessed among patients with ALC. Statistically significant mutual relationships are presented in Table 7.

Statistically notable relationships with respect to miR-1-3p were observed among ALC patients. This molecule correlated negatively with NEU (*p* < 0.05) and TGF-α (*p* < 0.05). A statistically significant negative relationship was also observed between miR-126-3p and AST (*p* < 0.05) and a positive relationship between miR-126-3p and miR-197-3p (*p* < 0.001). Moreover, the miR-197-3p molecule correlated notably with direct indicators of liver fibrosis—positively with PDF-AB (*p* < 0.05) and negatively with TGF-α (*p* < 0.01). The proposed diagnostic cut-off points, analysis of sensitivity and specificity, as well as PPV and NPV of the assessed indicators, are presented in Table 8.

The diagnostic value of the AUC regarding miR-126-3p molecule in the course of ALC was 0.568 and turned out to be statistically insignificant. The AUC value of the miR-197-3p molecule was 0.691 (*p* < 0.0001). Due to the significantly low serum expression of miR-197-3p molecule in both the research and control groups, it was not possible to determine its AUC value in ALC based on statistical analyses. Table 9 shows the results of the AUC values of examined markers in ALC patients. Figure 1 presents the ROC curve of miR-197-3p in patients with ALC.

## 4. Discussion

Following the literature on miRNA molecules over recent years, it seems that they already occupy a permanent place among potential markers of various diseases, as well as possible points of treatment. Nevertheless, the available data are often not clear enough to introduce miRNA molecules into the everyday practical canon of hepatology. At the same time, the search for new non-invasive indicators of liver fibrosis, including ALC, seems to be crucial. And from this perspective, the exploration of miRNA molecules as general indicators of oxidative stress, modulators of regeneration processes and oncogenesis seems to be fully justified in the context of liver pathologies. The expression of individual miRNA types is tissue-specific. From this perspective, miR-122-5p seems to be an example of a molecule found in significantly high concentrations within hepatocytes and, at the same time, in relatively small amounts in extrahepatic tissues. In mice, models lacking the gene encoding miR-122-5p, liver steatosis and fibrosis, as well as HCC, are developed [15]. Moreover, in the case of alcohol-related liver disease (ALD) or other forms of toxic liver damage, a significant increase in the concentration of this molecule was noted. On the other hand, the miR let-7a molecule turned out to be a significant predictor in a mice population model of alcoholic liver damage. Ethanol reduced the concentration of this molecule, promoting apoptosis by the activation of the Rb tumor suppressor gene (Rb) [16]. Yao J. and his research group confirmed the role of miR-132 and miR-29c in the course of ALD, demonstrating their direct impact on the regulation of genes involved in the course of ALD (SIRT1, FOXO1 and CDK1) [17]. Another study on miR-144 revealed its causative role in the stimulation of hepatic stellate cells and the promotion of liver fibrogenesis [18].

The function of miR-126-3p is directly related to modulating the course of oxidative stress, as indicated by the meta-analyses conducted so far. The highest concentrations of this molecule are observed mainly in epithelial cells, where miR-126-3p stimulates angiogenesis by inhibiting factors that act antagonistically to vascular endothelial growth factors [19]. Due to the direct relationship detected between miR-126-3p and oxidative stress, it seems reasonable to explore the potential role of this type of miRNA in hepatology. This molecule has not been significantly studied for its association with liver pathologies. So far, it has been established that it can regulate the function of the TICCR factor, which is directly involved, among others, in the development of HCC [20]. Additionally, studies on the development of liver cancer have shown the role of miR-126-3p as a transmitter of information between stellate cells and cancer cells, which promotes the development of liver fibrosis in the process of oncogenesis [21]. In turn, reduced expression of this molecule has been observed in cholangiocarcinoma [22]. In our study, the serum concentration of miR-126-3p among patients with ALC did not differ significantly compared to controls. Research results from the literature indicate that so far, there have been no attempts to correlate miR-126-3p expression with commonly known indirect and direct indicators of liver fibrosis or hematological markers used mainly in clinical trials. In our own research, among the mutual relationships, a positive correlation was observed between miR-126-3p and AST in the group of patients with ALC (*p* < 0.05). This seems to be the first observation of this type. In this case, it is difficult to formulate final conclusions based on the above relationship. Nevertheless, a direct relationship between miR-126-3p and the metabolic function of the liver was noticed in our own research.

So far, attempts to assess the role of the miR-197-3p molecule in liver pathologies have mainly allowed for the verification of its involvement in the course of HCC; however, the number of publications on this relationship is limited. Reduced expression of mi-R197-3p molecule in the course of this cancer, as well as the correlation of its lower concentration with a greater clinical and histopathological progression of the disease, were presented in the study conducted by Ni J. et al. [23]. Additionally, the above-mentioned researchers detected a relationship between higher miR-197-3p concentrations and the lower likelihood of metastases in the course of HCC, both in vitro and in vivo. Therefore, the role of this miRNA molecule turned out to be protective against the progression of cancer. In 2021, the results of a study dedicated to patients with metabolic dysfunction-associated steatotic liver disease (MASLD) were published [24]. The researchers undertook to verify the role of the HCGH18 gene in the course of the disease and its relationship with miR-197-3p. Increased activation of the mentioned gene and its negative correlation with the expression of miR-197-3p were found. In other observations dedicated to patients with MASLD, lower concentrations of this molecule were noted, as well as its association with the occurrence of liver fibrosis among a subgroup of patients with steatohepatitis. Other experiments have shown the potential impact of selected miRNA molecules (miR-340 and miR-200b-3p) on the inhibition or progression of the liver fibrosis process, respectively, through a direct effect on stellate cells. An independent indicator of this process was decreased or increased TGF-β1 concentration [25,26]. When analyzing the available literature, one may get the impression that there is an obvious lack of data regarding the potential relationship between miRNA molecules and TGF-α. Therefore, our own research bears the hallmarks of pioneering in this field of hepatology. Similar conclusions can be drawn in the context of the identified relationship between miR-197-3p and PDGF-AB. The issue of hematological markers in the context of liver diseases is another relatively new diagnostic area. So far, most attention has focused mainly on the following indicators: NLR and RPR, highlighting their potential role as markers of significant decompensation of liver cirrhosis. At the same time, researchers are looking at the potential role of PDW in liver pathologies. So far, higher values of this marker of PLT differentiation have been described in the course of LC caused by HBV infection and MASLD [27,28,29,30,31]. Additionally, in patients with alcoholic hepatitis, PDW value correlated positively with the MELD score.

The available literature data indicate that reduced expression of miR-1-3p accompanies many oncological diseases: the development of squamous cell carcinoma of the skin, prostate, urinary bladder, lungs, colon and HCC [32,33,34,35,36,37]. Additionally, clinical attempts to increase the expression of this molecule in an in vivo HCC model result in a decrease in tumor volume and attenuation of the process of intrinsic angiogenesis. In our study, among patients with ALC, no statistically significant results were observed in the context of miR-1-3p apart from its negative correlations with NEU (*p* < 0.05), TGF-a (*p* < 0.05) and laminin (*p* < 0.05). It is difficult to clearly comment on the relationship between miRNA and NEU. It appears to be the result of the inflammatory component of ALC. An important role here is played by the increased synthesis of interleukin 6 and TNF-α. At the same time, bacteria migrate from the gastrointestinal tract into the systemic circulation. Therefore, ALC patients often experience an increase in inflammatory markers. Due to the statistically significant relationship between miR-1-3p and NEU in our survey, it can be assumed that this molecule serves as an inflammatory marker in the course of ALC. In order to eliminate as much as possible the presence of additional factors influencing the number of NEUs in the blood of the studied patients, people with clinically significant infections—acute or chronic—were excluded. The statistically significant relationship between miR-3p and laminin identified in our own research opens a potentially new space in the diagnosis of liver fibrosis. Additionally, a significant correlation between the mentioned miRNA molecule and TGF-α was observed. The presented results suggest the validity of continuing work on the use of measurement of miR-1-3p expression in the course of ALC.

The existing relationship between serum miRNA expression and the process of liver fibrosis is indicated by statistically significant correlations between these molecules and direct markers of liver fibrosis noticed in our study. It is an important finding from the clinical perspective because such correlations prove the presence of participation of examined miRNAs in the process of liver fibrosis, and the results that were achieved bridge the gap in this aspect of hepatological diagnostics. Nevertheless, the role of miR-1-3p, miR-126-3p and miR-197-3p in the course of liver cirrhosis still requires further surveys. Lambrecht et al. created a model that included different molecules of miRNAs, which turned out to be superior in the diagnosis of significant liver fibrosis compared to FIB-4 and APRI [38]. Despite promising results, miRNAs are not used in everyday clinical practice in hepatology. The general idea of the current study was to explore the potential diagnostic role of miRNAs in cirrhotic patients together with an attempt to place them among already known former indices of liver fibrosis. Correlations between miRNAs and serological parameters of liver fibrosis were meant to be proof of the involvement of investigated miRNAs in the development of cirrhosis. At the current point of knowledge, our results still appear to require comprehensive confirmation, but they are simultaneously supported by statistically significant data. It is also worth mentioning that investigated miRNAs did not turn out to be superior to commonly known serological markers of liver fibrosis; nevertheless, our results suggest their notable involvement in this process.

On this basis, it seems advisable to evaluate these relationships in further studies, which may allow the isolation of miRNA types that, in everyday clinical practice, will act as independent and reliable markers of liver fibrosis in non-invasive diagnostics. Of note, in the current study, the miREIA technique was applied to assess the concentration of mi-RNAs; it allows for the assessment of certain subtypes of miRNAs. As was already emphasized, the choice of miR-1-3p, miR-126-3p and miR-197-3p in the current survey was directly related to the lack of data concerning the role of these molecules in the course of liver fibrosis. In the course of further work on this topic, it would also be extremely important to compare serum concentrations of miRNAs with their identical expression in liver biopsies. On the other hand, from an economic point of view, the miREIA technique could be used in the future in regular clinical settings, making the assessment of serological expression of miRNAs (after the validation of final results) quite commonly accessible.

The aspect of hematological markers also requires attention. In our research, no significant correlations were observed in patients with ALC, apart from a significant relationship between miR-1-3p and NEU. The AUC values of the tested miRNA molecules in the course of ALC are also important from a clinical point of view. Based on the results of our research, it can be concluded that the highest diagnostic value was observed for the miR- 197-3p (AUC = 0.691, *p* < 0.0001; suggested cut-off point < 1.01 amol/μL). In the case of miR-1-3p, the AUC value was not determined due to the very low serum expression of this molecule both in the research group and in the control group. Considering the AUC values of the miRNA molecules used in our own studies in the context of the AUC results of hematological markers and indirect and direct indicators of liver fibrosis, it can be assumed that the relatively highest diagnostic AUC value concerned hematological parameters and indirect indicators of liver fibrosis. Significantly lower AUC values (although at a comparable level) were characteristic of direct markers of liver fibrosis and the examined miRNA molecules. Despite these divergent results, the nature of which is difficult to explain even based on the latest literature, it seems that the various parameters cited above have significant relationships. This is proven by the correlations observed in the results of the current study. The attention was paid to significant relationships between the assessed miRNA molecules, hematological indices and direct markers of liver fibrosis. It is possible that the common denominator of these compounds is inflammation and the process of liver fibrosis itself, in which all the molecules mentioned above are involved. At the present stage of hepatological diagnostics, there is no clear data that would allow for a clear determination of the potential usefulness of the correlations observed in our results. Nevertheless, the innovative nature of the presented findings encourages further research in this area. More detailed analysis also requires determining the hypothetical reason for the lack of any significant relationships between the examined miRNA molecules and indirect indicators of liver fibrosis. It could be hypothesized that the observed significant correlation between miR-126-3p and AST in the ALC group should suggest the presence of other identical compounds between the assessed miRNA molecules and other indicators of hepatocyte function; however, no such relationships were observed.

It is worth noting here that despite the pioneering nature of the presented research, there are certain limitations and elements that should be given more attention in the future. The diagnosis of LC in our patients was based only on the result of a US examination. We did not perform liver biopsy or liver elastography with the use of Fibroscan or shear wave elastography. In further studies, it would be suitable not only to include imaging modalities in the assessment of liver fibrosis but also to correlate the serological expression of examined miRNAs with a degree of liver fibrosis. In our research, the serum expression of particular types of miRNAs was observed in patients with ALC. In parallel, a number of other non-invasive markers were assessed: direct and indirect indicators of liver fibrosis and hematological parameters. However, the primary goal was to assess the potential clinical usefulness of selected miRNA molecules.

## 5. Conclusions

Regardless of its limitations, the main finding of our study is a new insight into the relationships between serological expression of miRNAs and direct markers of liver fibrosis. The markings presented in our research were of a pilot nature. Based on the literature on the subject, these seem to be the first tests of this type carried out in the population of Polish patients with ALC.

## Figures and Tables

**Figure 1 biomedicines-12-02108-f001:**
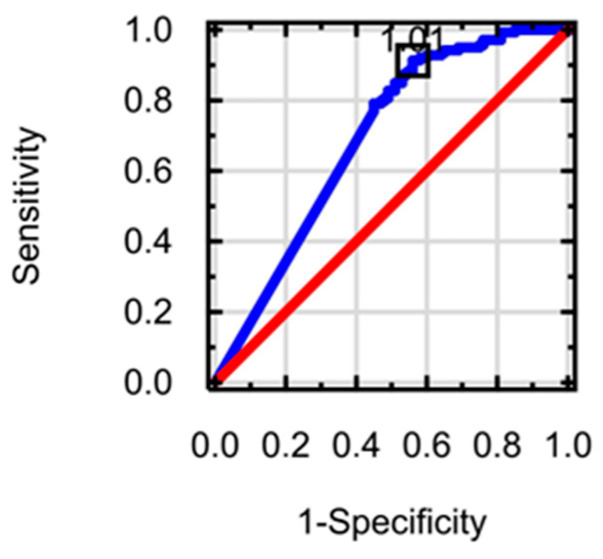
ROC curve (blue line) for miR-197-3p in the ALC group and random classifier (red line). AUC = 0.691, *p* < 0.0001. A proposed cut-off > 1.01 amol/µL.

**Table 1 biomedicines-12-02108-t001:** Clinical characteristics of people recruited to the study.

Parameter	ALC *n* = 139	Controls *n* = 100	Together *n* = 239
sex (m/w)	35/104	57/43	92/147
age (years) (x ± s; me; min-max)	54 ± 12; 55; 31–84	43 ± 15; 39; 20–85	52 ± 15; 54; 20–90
BMI (kg/m^2^) (x ± s; me; min-max)	22.68 ± 3.19; 22.65; 16.48–29.94	22.89 ± 2.38; 23.45; 16.18–36.99	-
diabetes *	0/139	-	-
arterial hypertension	35/139	-	-

* Negative results of confirmatory tests for diabetes: a fasting plasma glucose test, oral glucose tolerance test, random plasma glucose test and hemoglobin A1c test.

**Table 2 biomedicines-12-02108-t002:** Biochemical specifications of people included in the study and scoring of the scales used in the research group.

Parameter [Reference Range]	ALC					Controls				
	*x*	s	me	min	max	*x*	s	me	min	max
albumin [3.2–4.8 g/dL]	2.79	0.59	2.7 ****	1.35	4.65	4.05	0.45	4.09	3.17	4.8
bilirubin [0.3–1.2 g/dL]	6.17	7.58	2.6 ****	0.2	30.9	0.66	0.27	0.7	0.2	1.2
creatinine [0.5–1.1 g/dL]	0.83	0.19	0.8 **	0.5	1.1	0.81	0.81	0.7	0.5	1
INR [0.8–1.2]	1.6	0.59	1.41 ****	0.82	4.96	1.06	0.11	1.08	0.8	1.2
PT [10.4–13 s]	18.14	6.96	15.7 ****	9.2	56.1	11.67	0.8	11.7	10	13
AST [<34 IU/L]	103	91	77 ****	15	554	23	7	22	9	33
ALT [<31 IU/L]	52	55	37 ****	5	545	22	8	21	7	30
GGTP [<31 IU/L]	387	662	188 ****	16	5418	21	6	21	7	30
MELD	17	8	16	6	45	-	-	-	-	-

** *p* < 0.01, **** *p* < 0.0001.

**Table 3 biomedicines-12-02108-t003:** Results of tested microRNA molecules among study participants.

Parameter [Reference Range]	ALC					Controls				
	*x*	s	me	min	max	*x*	s	me	min	max
miR-126-3p [amol/μL]	0.81	1.527	0.06	0	9.69	0.88	1.22	0.38	0	5.83
miR-197-3p [amol/μL]	0.35	0.964	0.00 ****	0	5.18	1.95	2.67	0.39	0	12.7
miR-1-3p [amol/μL]	0.01	0.124	0	0	1.06	0.08	0.78	0	0	7.45

**** *p* < 0.0001.

**Table 4 biomedicines-12-02108-t004:** Results of evaluated hematological indices in the study groups.

Parameter [Reference Range]	ALC					Controls				
	*x*	s	me	min	max	*x*	s	me	min	max
PLT [130–400 × 10^9^/L]	118	81	92 ****	4	459	286	61	288	160	400
MPV [8–11fl]	9.11	1.5	8.7	6.6	14.4	9.18	0.88	9	8	11
MPR	0.15	0.3	0.09 ****	0.02	3.28	0.03	0.01	0.03	0.02	0.06
PDW [40–60%]	58.92	11.38	60.5 ****	20.4	95.9	50.8	5.63	51.2	40.1	60
PCT [0.12–0.3%]	0.13	0.09	0.1 ****	0	0.57	0.21	0.05	0.21	0.12	0.3
RDW [11–15%]	17.27	3.17	16.7 ****	12.2	27.9	13.38	1.1	13.45	11	15
RPR	0.26	0.38	0.18 ****	0.04	3.55	0.05	0.01	0.05	0.03	0.08
NEU [2.5–5 × 10^3^/μL]	5.88	4.8	4.48 **	1.04	30.4	3.71	1.13	3.66	2.5	4.9
LYM [1.5–3.5 × 10^3^/μL]	1.23	1.05	1.08 ****	0.26	11.4	2.3	0.69	2.26	1.5	3.48
NLR	6.27	7.03	4.07 ****	0.53	49.84	1.78	0.94	1.58	0.81	6.2
PLR	119.77	86.46	98.98 ****	0.7	435.82	138.65	60.42	125.09	55.94	327.27
RLR	19.23	12.34	15.68 ****	1.61	79.23	6.44	2.29	5.88	3.43	12.5

** *p* < 0.01, **** *p* < 0.0001.

**Table 5 biomedicines-12-02108-t005:** Results of determined indirect indices of liver fibrosis in the study groups.

Parameter	ALC					Controls				
	*x*	s	me	min	max	*x*	s	me	min	max
AAR	2.21	1.16	1.89 ****	0.18	7.57	1.12	0.38	1.09	0.43	2.86
APRI	4.43	7.08	2.53 ****	0.15	68.38	0.25	0.12	0.23	0.11	0.86
FIB-4	11.86	25.7	6.39 ****	0.69	287.59	0.81	0.48	0.72	0.23	3.27
GPR	16	28.78	6.83 ****	0.18	188.71	0.25	0.09	0.24	0.06	0.63

**** *p* < 0.0001.

**Table 6 biomedicines-12-02108-t006:** Results of serum direct markers of liver fibrosis examined among study participants.

Parameter	ALC					Controls				
	*x*	s	me	min	max	*x*	s	me	min	max
PICP (ng/mL)	62.95	31.53	59.91	6.15	161.12	58.26	37.39	44.18	0	202.89
PIIINP (ng/mL)	9.3	4.37	8.44 **	2.43	28.65	11.07	5.61	10.25	4.35	43.63
PDGF-AB (pg/mL)	18,310.19	8096.22	17,357.03 ***	1925.68	42,823.84	23,579.28	10,068.80	25,623.2	1638.2	47,758.70
TGF-α (pg/mL)	23.54	45.72	13.42 ****	0.872	507.09	28.44	17.21	24.59	1.31	93.55
Laminin (ng/mL)	955.82	681.38	827.88	101.933	2990.77	718.24	386.1	663.27	140.88	1813.88

** *p* < 0.01, *** *p* < 0.001, **** *p* < 0.0001.

**Table 7 biomedicines-12-02108-t007:** Statistically significant correlations of the tested microRNA molecules in the ALC group.

ALC		
Pair of Markers	R Spearman	*p*
miR-126-3p and AST	−0.181	*
miR-126-3p and miR-197-3p	0.32	***
miR-197-3p and PDGF-AB	0.169	*
miR-197-3p and TGF-α	−0.219	**
miR-1-3p and NEU	−0.179	*
miR-1-3p and TGF-α	−0.186	*
miR-1-3p and laminin	−0.176	*

* *p* < 0.05,** *p* < 0.01, *** *p* < 0.001.

**Table 8 biomedicines-12-02108-t008:** Specificity, sensitivity, PPV and NPV of the examined parameters in the ALC group.

Parameter	ALC				
	Cut-off	Sensitivity [%]	Specificity [%]	PPV [%]	NPV [%]
miR-126-3p [amol/μL]	<0.31	61	56	66	51
miR-197-3p [amol/μL]	<1.01	91	44	69	79
miR-1-3p [amol/μL]	-	-	-	-	-
MPV [fl]	<7.9	20	100	100	50
MPR	>0.05	83	94	94	81
PDW [%]	>59.4	60	96	94	65
PCT [%]	>0.12	63	99	99	68
RDW [%]	>15.1	73	100	100	75
RPR	>0.08	86	99	99	84
NLR	>2.71	73	89	89	72
PLR	<70.46	35	96	91	54
RLR	>8.68	90	86	89	87
AAR	>0.55	98	99	99	97
APRI	>1.59	70	92	92	68
FIB-4	>0.44	94	98	98	91
GPR	>1.73	93	96	97	90
PICP (ng/mL)	>50.4	67	56	76	45
PIIINP (ng/mL)	<8.02	46	81	83	42
PDGF-AB (pg/mL)	<25,171.82	80	53	78	57
TGF-α (pg/mL)	<16.1	57	78	84	47
Laminin (ng/mL)	>836.24	50	74	79	42

**Table 9 biomedicines-12-02108-t009:** Comparison of AUC values of assessed markers in ALC group.

Parameter	ALC	
	Diagnostic Accuracy of the Marker	
	AUC	*p*
miR-126-3p	0.568	-
miR-197-3p	0.691	****
miR-1-3p	-	-
MPV	0.569	-
MPR	0.922	****
PDW	0.767	****
PCT	0.832	****
RDW	0.925	****
RPR	0.965	****
NLR	0.852	****
PLR	0.629	***
RLR	0.938	****
GPR	0.991	****
AAR	0.874	****
APRI	0.979	****
FIB-4	0.982	****
PICP	0.565	-
PIIINP	0.628	***
PDGF-AB	0.662	****
TGF-α	0.69	****
Laminin	0.581	*

* *p* < 0.05, *** *p* < 0.001, **** *p* < 0.0001, - *p* > 0.05.

## Data Availability

Data are contained within the article.

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
