# Peer review of "microRNAs and Other Serological Markers of Liver Fibrosis in Patients with Alcohol-Related Liver Cirrhosis"

_biomedicines, 2024, doi:10.3390/biomedicines12092108_

Round 1

Reviewer 1 Report

Comments and Suggestions for Authors

Dear authors

The paper is novel but there are some things that might be corrected:

- In the end of the introduction, there might be a sentence saying the aim of the manuscript.

- An important drawback of the study is considering liver cirrhosis based only in imaging, and not in fibroscan of liver biopsy.

- The text is too big and all the sentences are followed without paragraphs, which makes more difficult to read the article. Please try to summarize it, concise all the information to be more clear for the readers.

- It is also very important to write an appropriate conclusion, indicating the main findings of your research.

Comments on the Quality of English Language

English language is good.

Author Response

Dear authors, The paper is novel but there are some things that might be corrected:

- In the end of the introduction, there might be a sentence saying the aim of the manuscript.

The additional sentence including the aim of the study was placed at the end of introduction.

- An important drawback of the study is considering liver cirrhosis based only in imaging, and not in fibroscan of liver biopsy.

Indeed, we perceive the lack of the assessment of liver fibrosis without dedicated imaging tools as a significant disadvantage of the survey. Nevertheless, it was a pilot study and we mainly focused on the evaluation of selected miRNAs in the general group of patients with liver cirrhosis - without looking for correlations between expression of miRNAs and the degree of fibrosis. Undoubtedly, we will include these data in our future papers. Moreover, we placed a comment to this drawback in the new paragraph of discussion, presenting limitations.

- The text is too big and all the sentences are followed without paragraphs, which makes more difficult to read the article. Please try to summarize it, concise all the information to be more clear for the readers.

According to your valuable suggestion, blocks of text (e.g. methodology and discussion) were divided into paragraphs.

- It is also very important to write an appropriate conclusion, indicating the main findings of your research.

The section of conclusions was added.

Reviewer 2 Report

Comments and Suggestions for Authors

The present report investigates the potential of microRNAs (miRNAs) and blood-test markers as non-invasive indicators of liver fibrosis in patients with alcoholic cirrhosis. The study found that serum miR-197-3p was significantly lower in alcoholic cirrhosis patients compared to the control group. Additionally, several blood-test parameters, including PLT, LYM, and RDW, were significantly altered in patients with alcoholic cirrhosis. This study also tried correlating miRNAs with indirect and direct blood-test markers liver fibrosis. The study concludes that while miRNAs show promise as non-invasive diagnostic tools for liver fibrosis, as miRNAs being novel biomarkers and therapeutic targets due to their involving gene regulation and cellular processes, future studies are needed to build and examine their clinical utility.  The context of the biological plausibility, gaps, novelty, miRNAs in relation to alcoholic cirrhosis are addressed.

Concerns:

- The authors should provide a clear justification for their selection of the target miRNAs

- Failed to correlated miRNAs with liver tissue markers in liver fibrogenesis

- This study relied on serum miRNA expressions without comparing them to miRNA expressions in liver tissue

- Additional patient recruitment and exclusion criteria as typically seen in clinical study reports are lacking

- Several of the study results are supposed to be typically presented in Results section

- Oversimplicity was noted in Statistic Analysis section-- only a basic overview of the statistical tests used is seen, while lacking typical details on specific software, versions, and settings used for data analysis

- Oversimplicity was also noted in Results section in which the transition requires to be streamlined and coherent in results shown

- The discussions lack depth and critical analysis in terms of mechanistic causality, relations, experimental and clinical implications, whatever

- Comparisons with known publications as typically seen in well-written reports are weak in the present discussions 

Author Response

The present report investigates the potential of microRNAs (miRNAs) and blood-test markers as non-invasive indicators of liver fibrosis in patients with alcoholic cirrhosis. The study found that serum miR-197-3p was significantly lower in alcoholic cirrhosis patients compared to the control group. Additionally, several blood-test parameters, including PLT, LYM, and RDW, were significantly altered in patients with alcoholic cirrhosis. This study also tried correlating miRNAs with indirect and direct blood-test markers liver fibrosis. The study concludes that while miRNAs show promise as non-invasive diagnostic tools for liver fibrosis, as miRNAs being novel biomarkers and therapeutic targets due to their involving gene regulation and cellular processes, future studies are needed to build and examine their clinical utility.  The context of the biological plausibility, gaps, novelty, miRNAs in relation to alcoholic cirrhosis are addressed.

 Concerns:

- 1. The authors should provide a clear justification for their selection of the target miRNAs

This additional information was added to the section of introduction.

- 2. Failed to correlated miRNAs with liver tissue markers in liver fibrogenesis

As we mentioned in the limitations of our survey, in future we plan to perform a similar study with not only serological assessment of miRNAs and direct markers of liver fibrosis, but also with their evaluation in liver biopsy specimens. It was the pilot study, therefore we focused only on the results obtained from the blood of study participants.

- 3. This study relied on serum miRNA expressions without comparing them to miRNA expressions in liver tissue

As we mentioned in the limitations of our survey, in future we plan to perform a similar study with not only serological assessment of miRNAs and direct markers of liver fibrosis, but also with their evaluation in liver biopsy specimens. It was the pilot study, therefore we focused only on the results obtained from the blood of study participants.

- 4. Additional patient recruitment and exclusion criteria as typically seen in clinical study reports are lacking

The section containing inclusion and exclusion criteria was modified.

- 5. Several of the study results are supposed to be typically presented in Results section

The missing data in the section of results were completed.

- 6. Oversimplicity was noted in Statistic Analysis section-- only a basic overview of the statistical tests used is seen, while lacking typical details on specific software, versions, and settings used for data analysis

The description of statistical analysis was improved.

- 7. Oversimplicity was also noted in Results section in which the transition requires to be streamlined and coherent in results shown

The section of results was modified.

- 8. The discussions lack depth and critical analysis in terms of mechanistic causality, relations, experimental and clinical implications, whatever

Thank you for this valuable comment. We tried to modify all of these aspects of discussion to make it more comprehensive.

- 9. Comparisons with known publications as typically seen in well-written reports are weak in the present discussions 

We tried to discuss other studies belonging to this field of hepatology, nevertheless the number of existing publications referring to some kind of correlations between miRNAs and markers of liver fibrosis among cirrhotic patients is limited. But after a subsequent research we included two more publications.

Reviewer 3 Report

Comments and Suggestions for Authors

In this study, the authors examined in ALC patients, the expressions of specific miRNAs and some of the new and established markers/parameters used in the diagnosis of fibrosis/cirrhosis. Correlations are studied between the expressions of these miRNA and some of the markers/parameters. Ample data is reported here, which indicates a substantial amount of work. These efforts need to be appreciated.

However, there are some concerns. 3 main areas where the study needs work are rationale, novelty and purpose.  

1.       One concern about the manuscript is the rationale for this study.  The authors have not stated the rationale in the study in the Abstract. Stating that- non-invasive indicators of liver fibrosis are still of importance, does not effectively justify this large study for which a lot of effort has been made, time and resources spent, ethics applied for. In the Abstract, the authors do mention the objective of the study. However, the aim- why do the authors wish to assess miRNA expression and corelate this with existing biomarker set is not explained.

There is an attempt to justify the study at the start of the Introduction (e.g line 38), line 43, and the Discussion but it is does not come across strongly.

2.       There are several miRNAs involved with liver physiology and pathology. Why only these miRNAs have been chosen for analysis? (miR-126-3p, miR-197-3p and miR-1-3p). Three is some explanation given in the Discussion. That needs to be brought into the Introduction section and strengthened.

3.       In the Abstract and in the Methods (line 57): Who are the control group? non-drinkers or drinkers who have not developed cirrhosis? The control group is extremely important and needs to properly defined.

4.       The Abstract and the Result section mentions about the corelation of these miRNAs with the other markers? How does this help clinically? what is the impact of this work at clinical level? What are the current loopholes that this study fills? Or how does this help current diagnosis? What purpose does this study serve? These questions need to be answered in relevant places of the document.

5.       Note that as it stands, this work is great from the research perspective where we try to understand the corelation between cell metabolites/proteins/ biomarkers in a certain pathological state, However, how useful this study is in regular clinical settings? needs to be justified, as it involves quality assurance validation, patents, costs etc.

6.       Methods section- Ethics is stated here. However, it seems like randomly sandwiched between different pieces of biological information. Please write this in a separate para. What was done with the patients who were diagnosed with ALC?

7.       The same is the situation with statistics, which is all written within that 1 big block of text.  Please separate into another para for clarity.

8.       The discussion is written as 1 big block of text. Please separate into paras that explain different concepts.

9.       I understand that the authors tested microRNAs. However, the results of most other parameters e.g. results on Table 4 or Table 5 e.g FIB-5 that they tested (with the aim of studying correlations) are already known for cirrhosis patients. So there is not much novelty in that context. Indeed, the correlations between these parameters and the miRNAs were not known, and the study does present that, but what value or how much value does it add to the current diagnosis approaches remains to be answered.  

Author Response

In this study, the authors examined in ALC patients, the expressions of specific miRNAs and some of the new and established markers/parameters used in the diagnosis of fibrosis/cirrhosis. Correlations are studied between the expressions of these miRNA and some of the markers/parameters. Ample data is reported here, which indicates a substantial amount of work. These efforts need to be appreciated. However, there are some concerns. 3 main areas where the study needs work are rationale, novelty and purpose.  

1.       One concern about the manuscript is the rationale for this study.  The authors have not stated the rationale in the study in the Abstract. Stating that- non-invasive indicators of liver fibrosis are still of importance, does not effectively justify this large study for which a lot of effort has been made, time and resources spent, ethics applied for. In the Abstract, the authors do mention the objective of the study. However, the aim- why do the authors wish to assess miRNA expression and corelate this with existing biomarker set is not explained.

There is an attempt to justify the study at the start of the Introduction (e.g line 38), line 43, and the Discussion but it is does not come across strongly.

Thank you for this comment. Indeed, we did not state it clearly, why did we decide to perform exactly such a study. We tried to improve this aspect of the manuscript. The abstract and the section of introduction were modified in order to emphasize the idea of the current paper. 

2.       There are several miRNAs involved with liver physiology and pathology. Why only these miRNAs have been chosen for analysis? (miR-126-3p, miR-197-3p and miR-1-3p). Three is some explanation given in the Discussion. That needs to be brought into the Introduction section and strengthened.

The reason for the choice of the examined type of miRNAs (miR-1-3p, miR-126-3p and miR-197-3p) was additionally explained in the introduction.

3.       In the Abstract and in the Methods (line 57): Who are the control group? non-drinkers or drinkers who have not developed cirrhosis? The control group is extremely important and needs to properly defined.

The definition of control group in the abstract and in the section of material and methods was improved. These were healthy non-drinkers with no identified liver disorders

4.       The Abstract and the Result section mentions about the corelation of these miRNAs with the other markers? How does this help clinically? what is the impact of this work at clinical level? What are the current loopholes that this study fills? Or how does this help current diagnosis? What purpose does this study serve? These questions need to be answered in relevant places of the document.

It is an important finding from the clinical perspective, because such correlations prove the presence of participation of examined miRNAs in the process of liver fibrosis and achieved results cover the loophole in this aspect of hepatological diagnostics. Nevertheless, a certain role of miR-1-3p, miR-126-3p and miR-197-3p in the course of liver cirrhosis still requires further surveys. The general idea of the current study was to explore a potential diagnostic role of miRNAs in cirrhotic patients together with an attempt to place them among already known, former indices of liver fibrosis. Correlations between miRNAs and serological parameters of liver fibrosis were meant to be the proof of the involvement of investigated miRNAs in the development of cirrhosis. At the current point of knowledge our results appear to still sound like novel speculations that require a comprehensive confirmation, but they are simultaneously supported by statistically significant data. As presented above, the context of observed correlations was described in a more detailed way in the discussion of the manuscript.  

5.       Note that as it stands, this work is great from the research perspective where we try to understand the corelation between cell metabolites/proteins/ biomarkers in a certain pathological state, However, how useful this study is in regular clinical settings? needs to be justified, as it involves quality assurance validation, patents, costs etc.

The issue of miRNAs together with everyday clinical settings was raised and the possibility of the practical use of tests used in the current manuscript was mentioned in the discussion. 

6.       Methods section- Ethics is stated here. However, it seems like randomly sandwiched between different pieces of biological information. Please write this in a separate para. What was done with the patients who were diagnosed with ALC?

Ethical information was written as a separate paragraph in the section of materials and methods. The patients with ALC qualified for the study were already under healthcare (outpatient clinics), thus they continued their treatment and after the inclusion to our study they underwent a single blood collection. 

7.       The same is the situation with statistics, which is all written within that 1 big block of text.  Please separate into another para for clarity.

The section of statistics was separated from the main text.

8.       The discussion is written as 1 big block of text. Please separate into paras that explain different concepts.

Thank you for this suggestion. The discussion required the edition, indeed. Its construction was remodeled and based on independent paragraphs.

9.       I understand that the authors tested microRNAs. However, the results of most other parameters e.g. results on Table 4 or Table 5 e.g FIB-5 that they tested (with the aim of studying correlations) are already known for cirrhosis patients. So there is not much novelty in that context. Indeed, the correlations between these parameters and the miRNAs were not known, and the study does present that, but what value or how much value does it add to the current diagnosis approaches remains to be answered.  

Our major aim was to discuss three types of miRNAs, nevertheless, we also wanted to evaluate their potential relationships with already known hepatic markers. It appears that such correlations were not investigated, so far. We tried to additionally present our point of view on this topic in the manuscript.

Reviewer 4 Report

Comments and Suggestions for Authors

Authors showed the results of miR analyses for assessment of liver fibrosis in alcohol-related liver cirrhosis (ALC). As authors mentioned, the assessment of liver fibrosis noninvasively is clinically important. Several issues remained to be addressed in present study.

1. In introduction section, authors should describe the significance for assessment of liver fibrosis in ALC patients. The disease burden or difficulty of liver fibrosis assessment should be described with references.

2. In present study, diabetes was not found in ALC patients. Authors should mention the definition of DM.

3. The results of US elastography were preferrable for assessment of the degree of liver fibrosis.

4. In present study, portal vein diameter >13 mm was defined as portal hypertension. It is not commonly used. 

5. In present study, miR was not superior than other markers. 

6. The amount of alcohol drinking should be also included to the variables. 

Comments on the Quality of English Language

Grammatical errors and spelling errors were found.

Author Response

Authors showed the results of miR analyses for assessment of liver fibrosis in alcohol-related liver cirrhosis (ALC). As authors mentioned, the assessment of liver fibrosis noninvasively is clinically important. Several issues remained to be addressed in present study.

1. In introduction section, authors should describe the significance for assessment of liver fibrosis in ALC patients. The disease burden or difficulty of liver fibrosis assessment should be described with references.

Thank you for this suggestion. We did not include basic information regarding liver cirrhosis and its monitoring, which is crucial in the context of the whole manuscript. We added these data at the beginning of the paper.

2. In present study, diabetes was not found in ALC patients. Authors should mention the definition of DM.

DM was defined under the table with features of persons included in the survey.

3. The results of US elastography were preferrable for assessment of the degree of liver fibrosis.
 We additionally add this context into limitations of the study. we would like to broaden the spectrum of the current investigation in further surveys with a more precise description of the severity of liver fibrosis among the patients - withthe use of elastography.

4. In present study, portal vein diameter >13 mm was defined as portal hypertension. It is not commonly used. 

Indeed, according to the literature, the diameter of portal vein in the US examination ranges between 7-13 mm, but may vary up to 15 mm. In the current study we did not perform any additional imaging studies except abdominal US examination. Our aim was to exclude prehepatic or posthepatic portal hypertension. In order to not to miss some kind of ongoing pathology in the portal hypertension, we chose the diameter of the portal vein of 13 mm as the threshold for our cirrhotic patients included in the study. Moreover, in the history of enrolled patients, there was no additional information about previous disorders related to the portal circulation. Our patients suffered only from the hepatic type of portal hypertension.

5. In present study, miR was not superior than other markers. 

Indeed, we did not find miRNAs to have a better diagnostic accuracy in comparison to other, already known markers of liver fibrosis. Nevertheless, the results suggest the involvement of these particles in the process of liver fibrosis, which was emphasized in the discussion.

6. The amount of alcohol drinking should be also included to the variables. 

We did not collect certain data on the exact volume of alcohol consumed by participants. Thus, we used only this one threshold: alcohol-related cirrhosis was diagnosed in the situation of a confirmed daily consumption of more than 20 g of pure alcohol in the case of women and 30 g - in the case of men. According to the controls: patients reported occasional alcohol consumption of no more than 10 g of pure alcohol per day or declared complete abstinence.

Round 2

Reviewer 1 Report

Comments and Suggestions for Authors

Dear authors

I am satisfied after some improvements.

Author Response

I am satisfied after some improvements.

Thank you for your valuable suggestions.

Reviewer 2 Report

Comments and Suggestions for Authors

Concerns:

- To enhance the description in Methods, the authors could consider addressing the following: 1. the version and relevant settings about Statistica 13; 2. how the cut-off values for miRNAs were statistically determined; 3. any other statistical methods used; 4. the methods used in AUC comparisons.

Comments on the Quality of English Language

Some phrases, like "cover the loophole" and "novel speculations," could be replaced with more precise and scientific text. The tone and style could also be revised to be concise, confident and assertive, given the statistical significance of the findings.

Author Response

To enhance the description in Methods, the authors could consider addressing the following: 

1. the version and relevant settings about Statistica 13;

Statistical analysis was performed with Statistica  version 13.3 (TIBCO Software Inc.) software for Windows. Relevant settings were not applied; calculation was done with this software. ROC analysis was performed using Medical Bundle for Statistica software. Optimal cut-off values for miRNA levels were determined using tangent method. This addidtional decription of statistical analysis was also included in the main text

2. how the cut-off values for miRNAs were statistically determined; 

Optimal cut-off values for miRNA levels were determined using tangent method.

3. any other statistical methods used; 

We added new sentences (from point 1. of the response to this review) to Statistical analysis, the part of "Materials and Methods" section.

4. the methods used in AUC comparisons.

AUC and respective p values were calculated using Medical Bundle for Statistica version 13.3.

Some phrases, like "cover the loophole" and "novel speculations," could be replaced with more precise and scientific text. The tone and style could also be revised to be concise, confident and assertive, given the statistical significance of the findings.

English language, concerning the style of the paper and inapprioprate expressions, was improved.

Reviewer 3 Report

Comments and Suggestions for Authors

The authors have addressed the raised concerns 

Author Response

The authors have addressed the raised concerns 

Thank you for your valuable comments.

Reviewer 4 Report

Comments and Suggestions for Authors

Revised manuscript was well-addressed to the reviewer's comments although some limitations as described in the discussion section or about liver fibrosis assessment were found. 

Comments on the Quality of English Language

English was well-written. Minor spelling and grammatical check are preferred. 

Author Response

Revised manuscript was well-addressed to the reviewer's comments although some limitations as described in the discussion section or about liver fibrosis assessment were found. 

Indeed, our study includes several limitations, nevertheless it was the first attempt to perform such a survey comparing diagnostic accuracy of selected miRNAs and already known markers of liver fibrosis. In further investigations we will assess the status of liver fibrosis in examined patients more adequately. Even though, achieved results in the current analysis appear to be reliable and trustworthy. Nevertheless they need to be verified in subsequent investigations.

English was well-written. Minor spelling and grammatical check are preferred. 

English language was improved.